# Michelangelo Effect in Cognitive Rehabilitation: Using Art in a Digital Visuospatial Memory Task

**DOI:** 10.3390/brainsci14050479

**Published:** 2024-05-09

**Authors:** Claudia Salera, Chiara Capua, Domenico De Angelis, Paola Coiro, Vincenzo Venturiero, Anna Savo, Franco Marinozzi, Fabiano Bini, Stefano Paolucci, Gabriella Antonucci, Marco Iosa

**Affiliations:** 1IRCCS Santa Lucia Foundation, 00143 Rome, Italy; c.salera@hsantalucia.it (C.S.); d.deangelis@hsantalucia.it (D.D.A.); p.coiro@hsantalucia.it (P.C.); v.venturiero@hsantalucia.it (V.V.); a.savo@hsantalucia.it (A.S.); s.paolucci@hsantalucia.it (S.P.); gabriella.antonucci@uniroma1.it (G.A.); 2Department of Mechanical and Aerospace Engineering, Sapienza University of Rome, 00184 Rome, Italy; capua.1697720@studenti.uniroma1.it (C.C.); franco.marinozzi@uniroma1.it (F.M.); fabiano.bini@uniroma1.it (F.B.); 3Department of Psychology, Sapienza University of Rome, 00185 Rome, Italy

**Keywords:** art therapy, neuroesthetics, neurorehabilitation, psychology

## Abstract

The Michelangelo effect is a phenomenon that shows a reduction in perceived effort and an improvement in performance among both healthy subjects and patients when completing a motor task related to artistic stimuli, compared to performing the same task with non-artistic stimuli. It could contribute to the efficacy of art therapy in neurorehabilitation. In this study, the possible occurrence of this effect was tested in a cognitive task by asking 15 healthy subjects and 17 patients with a history of stroke to solve a digital version of the classical memory card game. Three different types of images were used in a randomized order: French cards, artistic portraits, and photos of famous people (to compensate for the possible effects of face recognition). Healthy subjects were involved to test the usability and the load demand of the developed system, reporting no statistically significant differences among the three sessions (*p* > 0.05). Conversely, patients had a better performance in terms of time (*p* = 0.014) and the number of attempts (*p* = 0.007) needed to complete the task in the presence of artistic stimuli, accompanied by a reduction in the perceived effort (*p* = 0.033). Furthermore, artistic stimuli, with respect to the other two types of images, seemed more associated with visuospatial control than linguistic functions.

## 1. Introduction

Since the beginning of the 21st century, there has been a major increase in research into the effects of the arts on health and wellbeing. The World Health Organization reported scientific evidence from a wide variety of studies using diverse methodologies about the potential impact of art on both mental and physical health [1]. This is mainly due to the aesthetic, cognitive, sensorial, perceptive, and emotive engagement that art could evocate in a person [1]. Some studies showed a wide brain arousal, even involving the motor cortex, during the observation of an artistic painting [2,3,4]. These activations during the observation of an artistic masterpiece could be exploited by art therapy. Art therapy has two conventional approaches: based on art fruition (such as viewing paintings or listening to music, usually using masterpieces) or based on art production (such as painting or playing an instrument, despite the products not being able to be properly defined as art) [1]. However, thanks to new technologies, the patient could be asked to actively produce an art product with the illusion of having generated a masterpiece. This is the case of a study in which the movements of the hand of patients with a history of stroke underwent sonification, transforming the kinematics in sound [5]. If the movements were physiologically harmonious, the acoustic feedback was a pleasant piece of music; conversely, a dystonic movement produced a distorted sound. Virtual reality was used in a protocol of art therapy based on paintings in which the patient had the illusion of being able to replicate an artistic masterpiece such as the Creation of Adam by Michelangelo or the Birth of Venus by Botticelli [6]. In this study it was found that, during virtual painting, patients perceived less fatigue and had a better kinematic performance when they had the illusion of generating an artistic painting in comparison to when they simply painted a virtual canvas: this outcome was called the “Michelangelo Effect”. In a follow-up study, it was proven that the effect was mainly related to the replication of the painting than to the beauty of the stimulus [7]. In a randomized controlled trial, patients with a history of stroke benefitted more from this virtual painting protocol based on the Michelangelo Effect than from conventional physical therapy [8].

Despite the hypothesis in these that the Michelangelo effect stemmed from increased cognitive engagement among patients, thus enhancing their motivation, this effect was more commonly utilized in neuromotor rehabilitation rather than cognitive rehabilitation. Consequently, there exists a gap in the literature regarding the potential harnessing of the Michelangelo effect in cognitive tasks and its usability in the cognitive rehabilitation of patients with a history of stroke. Cognitive rehabilitation following stroke often targets the recovery of memory, attention, and executive functions [9]. Recent protocols frequently incorporate technological devices such as tablets and computers equipped with specifically developed apps or software, often employing a gamified approach with serious exergames [10]. One of the most commonly used and straightforward rehabilitative serious games is the “memory card” game, wherein patients are tasked with remembering the positions of cards in order to match pairs, thereby facilitating visuospatial memory training. Memory card games have been frequently employed in previous studies focusing on cognitive therapy for patients with a history of stroke [9,11,12]. Additionally, the digital memory card game has been utilized for rehabilitation in cerebral palsy [13]. Both real and virtual cards can be utilized, with French-suited cards featuring the four suits: hearts, clubs, diamonds, and spades. In cases involving children, cards often feature images of animals or cartoons. Computerized cognitive training based on the memory card game was reported to be effective in relation to training individual cognitive functions such as visuospatial memory, attention, and language; furthermore, improvements were observed in the performance of neuropsychological tests overall [12].

The aims of this study were (1) to assess the usability of a digital memory card game using artistic pictures, with the intention of harnessing the Michelangelo effect, in a group of healthy subjects and (2) to compare the performance and cognitive load of patients affected by stroke using three different types of stimuli: traditional French cards, images of artistic masterpieces, and pictures of well-known television journalists. The inclusion of the last group of images aimed to counteract the potential effects of familiarity associated with famous paintings and the activation of specific brain circuits involved in face recognition, which may differ from those activated by abstract symbols of French-suited cards [7,14,15].

## 2. Materials and Methods

### 2.1. Memory-Based Serious Exergame

We developed, using the platform Interacty (Holge SIA, Riga, Latvia), a digital version of the “memory card” game to be performed on a touchscreen tablet (9.7 inches, resolution 2048 × 1536) similar to previous versions of digital memory card games [10,13]. Sixteen cards were used for each session placed on 4 rows and 4 columns. The game started with all the cards faced down and the player was instructed to turn over two cards by clicking on them. If the two cards showed the same picture, then the cards remained turned over, otherwise the cards returned to being faced down. The system recorded the time and the number of attempts needed to complete the task, which consisted of pairing all the pairs of cards. The three different versions of cards (each presented in separated sessions) consisted of French cards, pictures of famous TV news journalists of national channels, and portraits of characters from famous paintings. More specifically, the portraits represented Adam from the Creation of Adam (Michelangelo), Plato from the School of Athens (Raffaello), Venus from the Birth of Venus (Botticelli), Flora from the Primavera (Botticelli), Mona Lisa (Leonardo), the lady with an ermine (Leonardo), the girl with a pearl earring (Vermeer), and Bacchus (Caravaggio). The position of the images in each session was randomized.

### 2.2. Experiment 1: Usability of the System

Fifteen healthy adults (mean age: 27 ± 6 years, 9 females) without neurological, cognitive, or orthopedic diseases were enrolled in this first experiment among students and coworkers of our hospital. They were asked to complete the task with their preferred hand and using their glasses if needed.

They performed three sessions (session order was counterbalanced across subjects) of the digital game: French cards, photos of famous people (television journalists), and images of artist’s paintings related to the Renaissance (Figure 1).

Similarly to previous studies on computerized serious games developed for rehabilitation [6,16,17,18,19], a User Satisfaction Evaluation Questionnaire (USEQ) and Nasa Task Load Index (NASA-TLX) were administered to subjects after the completion of each session. Both scales test six domains of self-perception concerning the usability and the perceived load demand of the tool. In particular, USEQ has six questions (for example: “Did you enjoy your experience with the system?”), and responses can be provided using a five-point Likert Scale, from 1 to 5. The six items test the experienced enjoyment, perceived successful use, ability to control, clarity of information, discomfort, and self-perceived utility of the performed exercise for rehabilitation. According to the Likert Scale, the analyzed scores ranged from 1 (strongly disagree) to 5 (strongly agree). NASA-TLX has six questions with a ten-point numerical rating scale for each one of the items (for example: “How physically demanding was the task?”). It tests the self-perceived mental demand, physical demand, time pressure, perceived success, fatigue, and stress. The scores ranged from 0 (not at all) to 10 (the maximum possible).

### 2.3. Experiment 2: Tests on Patients

Seventeen patients with a history of stroke (mean age: 65 ± 15 years, 9 females, time from acute event: 23 ± 18 days, 11 with ischemic stroke, 11 with stroke in the right hemisphere) were enrolled in the second experiment of this study according to the following inclusion criteria: age ranging between 18 and 85 years, clinical diagnosis of stroke confirmed by computerized axial tomography or magnetic resonance imaging, a cognitive level adequate to understand the required instructions (Mini-Mental State Examination > 24), absence of unilateral spatial neglect, and absence of visual impairments that could not be corrected with glasses. All these patients had recovered in our rehabilitation hospital, and the medical staff identified them according to the aforementioned inclusion criteria. Patients were asked to complete the task with the less-affected hand and using their glasses if needed. Before the experiment with the tablet, patients were assessed using the tests of the Oxford Cognitive Screen [20]. The Italian version and the data interpretation reported by Iosa and colleagues [21] were used.

### 2.4. Statistical Analyses

Data are reported in terms of mean ± standard deviation. Repeated measures analysis of variance (RM-ANOVA) was used to compare the within-group measured variables among paintings, photos, and cards. Greenhouse–Geisser correction was used to compensate for any possible violations of the RM-ANOVA assumptions, and Tukey’s correction was applied to *p*-values for post-hoc tests. The correlation coefficient R was used to compute the association between variables. For patients, the correlation was also tested with the following clinical variables: items 3 and 5 of the MMSE, related to working memory and short-term memory (word repetition and word recall); and the tests of the OCS covering the following tasks (domains): picture naming (memory), picture pointing (visuospatial control), spatiotemporal location (orientation), sentence reading (language and arithmetic), number calculation (language and arithmetic), sentence recall (memory), and figure recall (memory). The alpha level of statistical significance associated to the rejection of the null hypothesis was fixed at 0.05 for all tests.

## 3. Results

### 3.1. Experiment 1: Usability of the System

Figure 2 shows the time taken and the attempts needed to complete each session of the memory task for healthy subjects. Neither time (F(2,14) = 0.04, *p* = 0.926, η_p_^2^ = 0.003) nor the number of attempts (F(2,14) = 0.23, *p* = 0.789, η_p_^2^ = 0.016) significantly varied among sessions.

Table 1 reports the scores of USEQ and NASA-TLX for healthy subjects. No significant differences were observed among the three sessions (*p* > 0.28 for all the items of USEQ and *p* > 0.14 for all the items of NASA-TLX). For all the three types of sessions, the results showed that the tasks were more mentally than physically demanding (*p* < 0.005).

### 3.2. Experiment 2: Tests on Patients

Also, patients reported higher levels of usability assessed by USEQ with a low discomfort. No significant differences were noted among the three sessions (see Table 2). In terms of NASA-TLX, as expected, patients required more time and attempts to complete the task; however, despite this, their USEQ scores were similar to those of healthy subjects, and the differences were not statistically significant for any of these scores. In terms of NASA-TLX, a statistically significant difference was observed in terms of the effort reported by patients, with less fatigue associated with completing the task with paintings compared to that reported for cards or photos of famous people (F(2,14) = 3.98, *p* = 0.033, η_p_^2^ = 0.199).

This reflected the difference measured among the three performances and is shown in Figure 3. Patients required less time (F(2,14) = 4.93, *p* = 0.014, η_p_^2^ = 0.236) and less attempts (F(2,14) = 7.86, *p* = 0.007, η_p_^2^ = 0.329) to complete the task with artistic pictures. Post-hoc tests highlighted that these results were mainly due to a difference in time between paintings and cards (*p* = 0.025, where no difference was observed between photos and cards: *p* = 0.410), and in number of attempts for both paintings and photos with respect to cards (*p* = 0.025 and *p* = 0.023, respectively).

### 3.3. Experiment 2: Correlations

During the French card session, patients showed a strong significant correlation between the time taken to complete the task and the relevant number of attempts (R = 0.826, *p* < 0.001). Both these performance parameters were also significantly correlated with episodic memory (*p* < 0.05; R = −0.755, R = −0.524, respectively), number calculation (R = −0.638, R = −0.599), and sentence reading (R = −0.574, R = −0.653). Then, the time taken to complete the task (but not the number of attempts) correlated with picture naming (R = −0.688), and the number of attempts (but not time taken) correlated with recall and recognition (R = −0.518). Finally, the perceived effort measured by NASA-TLX correlated with the memory item of the MMSE (R = −0.636, *p* = 0.006).

When artistic pictures were used instead of French cards, we did not find a significant correlation between the time and number of attempts needed to complete the task (R = 0.450, *p* = 0.070). According to this result, time (but not number of attempts) was found to be correlated with picture pointing (R = −0.511) and sentence reading (R = −0.665), whereas the number of attempts (but not time) was correlated with recall and recognition (R = −0.602) and the memory item of the MMSE (R = −0.604). The perceived effort was not correlated with any of the assessed parameters.

For photos of TV journalists, time and the number of attempts were significantly correlated (R = 0.792), and both these variables were correlated with the number calculation (R = −0.495, R = −0.506, respectively). The number of attempts was also correlated with recall and recognition (R = −0.483). The perceived effort was correlated with the second item of the MMSE (R = −0.512).

## 4. Discussion

First of all, both healthy subjects and patients judged the digital version of memory card as being highly usable. They also reported a low level of discomfort. These results were independent of the type of stimuli presented: French cards, photos of people, or artistic portraits. Positive judgments were also given in terms of NASA-TLX scores. Interestingly, only patients showed a significant difference among the three sessions in terms of the perceived effort. All parameters of the USEQ for both samples were not statistically affected by the type of stimulus, confirming that the usability of the system was not diminished by replacing images of French playing cards with those of artistic paintings or famous TV journalists. Similar results were obtained for the scores of NASA, although the fatigue perceived by patients was lower in the presence of artistic stimuli, which is consistent with the findings from previous studies on virtual paintings [7]. The reduction in the perceived effort, accompanied by a better performance in the presence of artistic stimuli, was previously defined as the Michelangelo effect [6], and used in rehabilitation protocols for subjects with a history of stroke [8]. The results of the present study did not show a “Michelangelo effect” for healthy subjects. In fact, their performance and their perceived effort were not significantly different among the three conditions. Previous studies also reported this effect in healthy subjects during virtual paintings [6] and virtual sculpturing [16]. This difference could be explained by the fact that the digital memory game task proposed in this study was easier than the tasks tested in virtual reality. The easiness of this task may have implied a “ceiling effect” on the judgments of the perceived effort. This interpretation is based on a previous study showing that memory tasks that are too simple for healthy adults are often affected by the ceiling effect [22]. In that study, this ceiling effect was less present in patients with a cortical degenerative condition [22].

Similarly, for the patients with a history of stroke enrolled in our study, the memory task was also not so simple, and the time needed to complete the task as well as the number of attempts were obviously longer than those recorded for healthy subjects. Interestingly, these two parameters were significantly lower when patients turned artistic images than when they performed the same task with French cards. Furthermore, they reported a lower perceived effort in the presence of artistic stimuli. These results are perfectly in line with the presence of a “Michelangelo effect” for patients with a history of stroke [6].

Correlations with clinical parameters revealed interesting results. For both the sessions with French cards and with photos of famous people, the time and the number of attempts needed to complete the task were significantly correlated with each other, and the perceived effort was related to the patient’s memory, assessed by the recall item of the MMSE. These correlations were not statistically significant in the presence of artistic stimuli. The absence of a correlation between time and number of attempts could be related to a less systematic approach by patients in the presence of paintings. As reported in the studies about the Michelangelo effect, in the presence of artistic images, the patients made less kinematic errors [6]. Then, the perceived effort was not significantly correlated with the patient’s memory, suggesting that the Michelangelo effect was generalized independently by the patient’s mnemonic capacity. On the other hand, this effect was observed for virtual paintings [6], virtual sculpturing [16], and now, for memory card gaming, suggesting that it could be present in very different tasks.

Regarding the assessed cognitive functions, we expected to find a significant correlation between the performance and the items related to memory.

The OCS-recall and recognition item results correlated with the number of attempts for all three types of images, suggesting that it was the most important cognitive function involved into this task. Since this OCS-item is related to verbal recall and recognition [20,23], although the task of the memory game was not verbal, it is possible that patients needed a verbalization in their mind.

The OCS-episodic memory item result correlated with the time taken and number of attempts only for cards. The score of the recall item of the MMSE, related to episodic memory, was found to be correlated with the perceived effort assessed by NASA-TLX during the task performed with the cards and the photos, but not for paintings, for which the correlation of this item was with the number of attempts.

Regarding the other cognitive functions, the score recorded for the number calculation item result correlated with the time taken and number of attempts for cards and photos, but not for paintings. It should be noted that the number calculation is assessed by OCS with the purpose of checking for preserved mathematical basic abilities (such as the recognition of the number reported on the cards, or counting the remaining cards to turn) and not to detect higher-level math deficits [20]. Significant correlation with the OCS-item of picture naming was found for cards and photos, but not for paintings. The OCS-item related to language, and in particular, sentence reading, was found to be correlated with the time taken and number of attempts for cards, and with the time taken for paintings.

Finally, the OCS-item picture pointing was found to be correlated with the time needed to complete the task only for paintings. This item was classified as being related to semantic cognitive function and, hence, to language domain in the original [20] and following versions [24,25,26] of the OCS, although a recent study suggested it be reclassified as being related to visuomotor control [21].

All these results seem to suggest the presence of the Michelangelo effect with artistic stimuli for patients with a history of stroke, with an improvement in their performance and a reduction in their perceived effort when they interacted with artistic stimuli. The recall of positions of paintings seemed to be associated with different cognitive functions with respect to the other two types of stimuli. In fact, despite memory recall and recognition functions for all types of images being important, the performance of patients in the sessions with cards and photos seemed to be related to the language and arithmetic domain. Language and numbers were two different domains of the original version of the OCS [20], but a recent principal component analysis revealed that they could be classified as a single domain [21]. The importance of this domain seemed to be limited to artistic stimuli, for which visuomotor control seemed to be crucial. This result could be interpreted as an involvement of linguistic and logical intelligence [27,28] for sessions with cards and photos, probably in terms of the need to verbalize the name of cards (number and sign) or the name or description of famous people reported in pictures. Conversely, visual processing and, hence, visual and kinesthetic intelligence [27,28] could be more involved in artistic stimuli.

The different involvement of cognitive domains, together with the possible engagement elicited by artistic stimuli reported in previous studies [1,29,30] could be at the basis of the observed Michelangelo effect in patients with a history of stroke during this memory task.

The results of this study should be carefully considered in light of its limitations. The first is the reduced sample size, especially for patients with a history of stroke, also given the possible heterogeneity of the cognitive deficits of this population [31]. Another limit is that we assessed a restricted part of the involved cognitive functions. For example, we did not assess the executive functions, that play a fundamental role in the rehabilitation of patients with a history of stroke [32]. Executive functions are a set of higher-order cognitive skills that support self-regulation and goal-directed behavior, comprising three main cores: inhibition, working memory, and cognitive flexibility, which in turn, contribute to reasoning, problem solving, and planning [33]. The OCS contains a switching task (in which the patient is asked to connect circles and squares from largest to smallest), used to assess executive functions. Because this task was originally categorized under the macro-domain of attention in the original OCS version [20], and being that it is more closely related to inhibition and cognitive flexibility than to working memory, we did not include it in our assessment. Further studies should investigate the potential role of executive functions in the Michelangelo effect. We also used two simple scales (USEQ and NASA-TLX) to assess the usability and load of the task. However, good face validity and reliability were demonstrated for the USEQ and NASA-TLX (Cronbach’s alpha > 0.7 for both of them) [34,35].

Finally, this study has a cross-sectional design, and a randomized controlled trial could be conducted to investigate the efficacy of cognitive neurorehabilitation designed to exploit the Michelangelo effect for artistic stimuli, as was carried out for motor neurorehabilitation [8].

## 5. Conclusions

This study also reports evidence for a Michelangelo effect with patients with a history of stroke when carrying out a cognitive task, particularly a memory task, in the presence of artistic stimuli. The usability of the developed digital memory card was rated highly by both healthy subjects and patients, with an adequate cognitive load for the task. These results also confirmed the effectiveness of the adopted methodologies in the development of the memory-based serious exergame. The Michelangelo effect, characterized by improved performance and reduced perception of fatigue in the presence of artistic stimuli, was observed in patients with a history of stroke. As reported by the World Health Organization, art therapy may induce benefits through the modulation of neurotransmitters, such as serotonin, reductions in stress hormones such as cortisol, and decreases in inflammatory immune responses, and contribute to enhancing emotional (e.g., self-expression, positive mood induction, and diversion) and cognitive aspects (e.g., stimulation of memory) [1,36]. Finally, our results may provide a first insight into a possible cognitive interpretation of the Michelangelo effect in accordance with the efficacy of art therapy protocols reported in the literature for cognitive treatments [1,37,38]. Further studies should investigate, through a randomized controlled trial, the efficacy of the developed serious exergame in leveraging the Michelangelo effect for the recovery of cognitive functions, especially memory, in patients affected by stroke.

## Figures and Tables

**Figure 1 brainsci-14-00479-f001:**
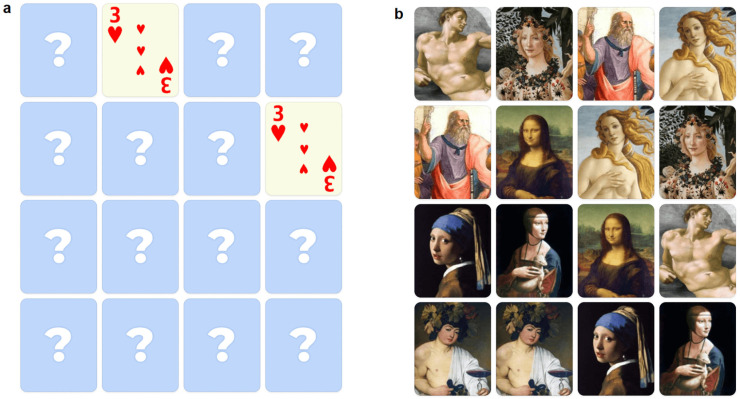
The digital memory game developed using the platform Interacty: (**a**) the version featuring French cards with a couple of them correctly matched and all other cards facing down; (**b**) the version featuring paintings, with all artworks turned face up to display the selected artworks.

**Figure 2 brainsci-14-00479-f002:**
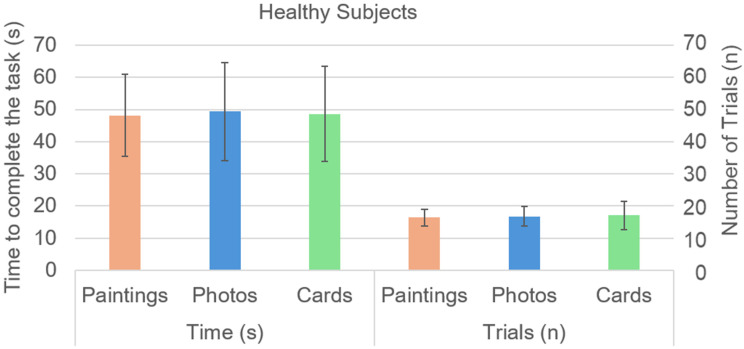
Mean (columns) and standard deviation (bars) of the time (in seconds, on the left) and the number of attempts (on the right) required to complete the task in the three sessions by healthy subjects: paintings (orange columns), photos (blue columns), and cards (green columns).

**Figure 3 brainsci-14-00479-f003:**
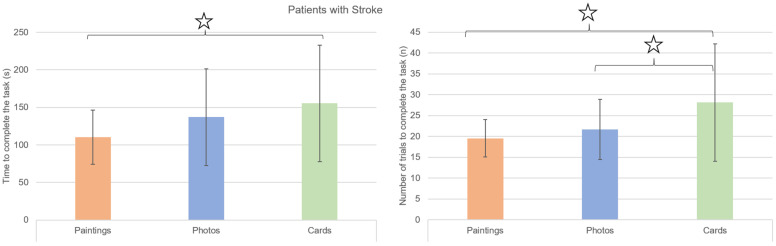
Mean (columns) and standard deviation (bars) of the time (in seconds, on the left) and the number of attempts (on the right) required to complete the task in the three sessions by patients with a history of stroke: paintings (orange columns), photos (blue columns), and cards (green columns). Stars highlight the statistically significant differences found by post-hoc tests.

**Table 1 brainsci-14-00479-t001:** Mean ± standard deviation of the items of USEQ and of NASA-TLX for healthy subjects, with F-values (and relevant degrees of freedom), *p*-values, and partial eta squared (η_p_^2^) as results of the RM-ANOVA.

Healthy Subjects		Paintings	Photos	Cards	F(2,14)	*p*-Value	η_p_^2^
USEQ	Experienced enjoyment	4.4 ± 1.1	4.1 ± 0.9	4.2 ± 0.7	1.20	0.315	0.079
	Successful use	4.9 ± 0.3	4.7 ± 0.6	4.7 ± 0.6	1.00	0.370	0.067
	Ability to control	4.6 ± 0.7	4.8 ± 0.6	4.6 ± 0.7	1.31	0.282	0.086
	Clarity of information	4.9 ± 0.3	4.9 ± 0.4	4.7 ± 0.8	0.68	0.440	0.047
	Discomfort	1.3 ± 0.8	1.3 ± 0.8	1.2 ± 0.8	1.00	0.381	0.067
	Perceived utility	4.3 ± 1.2	4.3 ± 1.1	4.4 ± 1.1	0.16	0.839	0.011
NASA-TLX	Mental demand	42.0 ± 20.6	40.3 ± 22.2	38.3 ± 25.0	0.46	0.621	0.032
	Physical demand	13.5 ± 17.7	15.6 ± 20.0	9.9 ± 10.5	1.39	0.263	0.090
	Temporal Demand	20.1 ± 19.2	27.2 ± 23.0	15.3 ± 14.8	2.11	0.148	0.131
	Satisfaction	64.4 ± 22.6	65.1 ± 21.0	62.0 ± 20.5	0.22	0.801	0.016
	Effort	40.0 ± 26.5	35.5 ± 21.6	41.2 ± 22.8	0.55	0.558	0.038
	Frustration	8.3 ± 9.5	8.9 ± 10.3	8.5 ± 9.7	0.10	0.853	0.007

**Table 2 brainsci-14-00479-t002:** Mean ± standard deviation of the items of USEQ and of NASA-TLX for patients, with F-values (and relevant degrees of freedom), *p*-values, and partial eta squared (η_p_^2^) as results of the RM-ANOVA.

Patients		Paintings	Photos	Cards	F(2,16)	*p*-Value	η_p_^2^
USEQ	Experienced enjoyment	4.8 ± 0.6	4.6 ± 0.6	4.6 ± 0.6	2.55	0.108	0.137
	Successful use	4.3 ± 1.0	4.4 ± 1.1	4.3 ± 1.2	0.08	0.794	0.005
	Ability to control	4.6 ± 0.8	4.5 ± 0.9	4.5 ± 0.9	2.13	0.135	0.118
	Clarity of information	4.9 ± 0.2	4.9 ± 0.2	4.9 ± 0.2	0.01	0.999	0.001
	Discomfort	1.0 ± 0.2	1.1 ± 0.3	1.2 ± 0.4	2.55	0.108	0.137
	Perceived utility	4.7 ± 0.6	4.6 ± 0.6	4.6 ± 0.6	2.13	0.135	0.118
NASA-TLX	Mental demand	22.8 ± 23.7	19 ± 25.5	25.3 ± 28.7	0.50	0.580	0.030
	Physical demand	4.3 ± 16.9	0.2 ± 0.4	0.2 ± 0.4	1.00	0.379	0.059
	Temporal Demand	5.4 ± 12.6	3.2 ± 6.5	2.5 ± 5.5	0.67	0.438	0.040
	Satisfaction	74.4 ± 27.8	74.7 ± 28.1	71.3 ± 29.5	0.56	0.536	0.034
	Effort	6.9 ± 16.9	17.5 ± 24.3	15.4 ± 22.8	3.98	0.033	0.199
	Frustration	0.8 ± 2.4	2.0 ± 5.2	4.2 ± 8.1	2.21	0.150	0.121

## Data Availability

Anonymous data are available on request to the corresponding author. The data are not publicly available due to clinical privacy issue.

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
