# Peer review of "Michelangelo Effect in Cognitive Rehabilitation: Using Art in a Digital Visuospatial Memory Task"

_brainsci, 2024, doi:10.3390/brainsci14050479_

Round 1

Reviewer 1 Report

Comments and Suggestions for Authors

An interesting work. Nevertheless, I do believe a more thorouht cognitive assesment is lacking, specially regarding executive functions. 

Reviewer 2 Report

Comments and Suggestions for Authors

See attachment

Comments on the Quality of English Language

See attachment

Reviewer 3 Report

Comments and Suggestions for Authors

1. The title suggests that the author intends to analyze the Michelangelo effect and the digital visuospatial memory task. It would be beneficial if the author could provide a more comprehensive theoretical explanation of the rationale for this analysis, its research value, and the problems it aims to address.

2. In addition, the author should provide more theoretical evidence for the beneficial effects of the digital visuospatial memory task in cognitive rehabilitation. The manuscript currently lacks substantial explanation or value in this regard. It would be helpful if the author could address this concern in more detail.

3. Towards the end of the introduction, the author mentions a memory card game for patient use. It would be helpful if the author could provide more theoretical or practical arguments to support this use and analyze its research value.

4. In the methods section, it would be beneficial if the author could provide more information about the reasons for using the experiment and the analytical procedures suggested by this experiment.

5. In addition, the author should provide more information about the theoretical explorations of the research tools used, as well as their validity and reliability.

6. In the results section, most of the statistical results did not show significance. It would be helpful if the author could discuss these results and analyze the implications or limitations these non-significant results may have on the study.

Round 2

Reviewer 2 Report

Comments and Suggestions for Authors

Comments:
This is the second round of this manuscript and this reviewer have read the manuscript and found that the authors have been addressed the changes carefully given by this reviewer and the manuscript has significantly improved. However, after evaluation of all changes in the manuscript this reviewer is willing to accept the manuscript for publication in Brain Sciences Journal. I appreciate the efforts of all authors during this research. In addition, the following are minor changes for the further improvement of this manuscript:
1.    In section 2.1, remove the https link and cite it by a proper reference.
2.    In introduction section, the last paragraph is regarding aims; write effective bullet points with research objectives to achieve the aims sparingly and strategically.
3.    Section 2.4 is regarding ethical approval and informed consent. Write this statement in the last of manuscript before references.
4.    In conclusion, the authors have still not concluded that the chosen methodologies produce good results. The real values of the outcome must be mentioned in a brief statement.
5.    Finally, write the conclusion as a separate section from discussion with future directions.

Comments on the Quality of English Language

Accepted with minor changes

Author Response

REVIEWER 2

This is the second round of this manuscript and this reviewer have read the manuscript and found that the authors have been addressed the changes carefully given by this reviewer and the manuscript has significantly improved. However, after evaluation of all changes in the manuscript this reviewer is willing to accept the manuscript for publication in Brain Sciences Journal. I appreciate the efforts of all authors during this research.

AUTHORS: Thank you for the positive judgment about our manuscript and for the following further comments helpful to improve the final version of our work

In addition, the following are minor changes for the further improvement of this manuscript:
1.    In section 2.1, remove the https link and cite it by a proper reference.

AUTHORS: We have removed the web address reporting the details of the company that developed this platform. The link was moved into the acknowledges section.

  1.  In introduction section, the last paragraph is regarding aims; write effective bullet points with research objectives to achieve the aims sparingly and strategically.

AUTHORS: We have rewritten the aims using numbered bullet points as follows:

“The aims of this study were: 1) to assess the usability of a digital memory card game using artistic pictures, with the intention of harnessing the Michelangelo effect, in a group of healthy subjects 2) to compare the performance and cognitive load of patients affected by stroke using three different types of stimuli: traditional French cards, images of artistic masterpieces, and pictures of well-known television journalists”.  

  1.  Section 2.4 is regarding ethical approval and informed consent. Write this statement in the last of manuscript before references.

AUTHORS: Done, we have now removed Section 2.4 and reported those information in the proper section at the end of the manuscript.

  1.  In conclusion, the authors have still not concluded that the chosen methodologies produce good results. The real values of the outcome must be mentioned in a brief statement.

AUTHORS: According to this comment, we have added the following sentence: “These results also confirmed the effectiveness of the adopted methodologies in the de-velopment of the memory serious exergame.”

  1.  Finally, write the conclusion as a separate section from discussion with future directions.

AUTHORS: Done, we have now defined a new paragraph titled “5. Conclusions”. We have also added into this section the following statement about future directions: “Further studies should investigate, through a randomized controlled trial, the efficacy of the developed serious exergame in leveraging the Michelangelo effect for the recovery of cognitive functions, especially memory, in patients affected by stroke.”

Reviewer 3 Report

Comments and Suggestions for Authors

The author made more revisions to this manuscript based on my concerns. I suggest this manuscript should be accepted in the present edition.

Author Response

Thank you for the positive judgment about our work